# Nesting Ecology of European Hedgehogs (*Erinaceus europaeus*) in Urban Areas in Southeast Spain: Nest Habitat Use and Characteristics

**DOI:** 10.3390/ani13152453

**Published:** 2023-07-29

**Authors:** Jana Marco-Tresserras, Germán M. López-Iborra

**Affiliations:** Department of Ecology, University of Alicante, 03690 Alicante, Spain; janamarco13@gmail.com

**Keywords:** day nest, resting sites, habitat selection, urban habitats, *Erinaceinae*

## Abstract

**Simple Summary:**

Nesting sites are one of the main factors driving the presence or absence of hedgehogs in an area, along with food availability and the presence of predators. However, little attention has been paid to them in the literature beyond the study of the hibernacula (winter nests) in northern latitudes, with only a few studies conducted in summer and none of those being in their southern range. Here, we analyse, for the first time in detail, the environmental characteristics of where hedgehogs chose to establish their nests in two urban parks in Alicante (Spain). Our results show that there is a sex difference in the structures selected for nesting. Males used a greater variety of structures than females, which were more selective and showed a preference for certain structures that offered a higher degree of shelter. Both sexes used a similar number of nests; however, the males’ nests were more widely dispersed, and they changed nests more often than females, who reused the same nest more frequently. The location of the nests was also influenced by the different topographies and management of both urban areas, stressing the importance of good management of urban green areas in the conservation of this species.

**Abstract:**

Appropriate nesting sites are needed for the presence of European hedgehogs (*Erinaceus europaeus*) in an area, along with food availability. However, little attention has been paid to them in the literature. This study aimed at analysing, for the first time, the environmental characteristics of nesting sites chosen by hedgehogs, their spatial distribution and the effect of sex and season on them in two types of urban parks in southeastern Spain. A total of 31 hedgehogs were equipped with GPS devices, and 130 hedgehog nests were located and described. Both sexes had a similar number of nests; however, the spatial distribution of the male nests was larger, and they changed nests more frequently than females. The environment around the nests and hosting structures used also differed between the sexes, with males using a higher variety of nesting structures available and females being more selective. The differences in topography and habitat composition of the two urban parks also affected hedgehog nesting ecology, especially in reference to artificial elements like cat feeders. Further studies of nest locations at a microhabitat level are needed to contribute to a better understanding of a hedgehog’s requirements, fostering the design of more effective conservation strategies.

## 1. Introduction

Species distribution is determined by a set of factors that are vital for the species’ survival. Among them, nest sites (understood as resting places not only for breeding), food availability and the presence of predators seem to be the main variables limiting the presence of European hedgehogs (*Erinaceus europaeus*) in an area [1,2,3,4,5]. Despite their vital importance, nests and refuges are not fully appreciated, and, as in other mammals, studies of hedgehogs have usually concentrated on other aspects of their biology and the behaviour of active animals [3].

The European hedgehog (hereafter ‘hedgehog’) is a medium-sized nocturnal insectivorous mammal found naturally from the Iberia Peninsula northwards into Scandinavia. Although timings may differ in relation to climate, sex, body size and condition, hedgehogs typically hibernate between November and April, making winter nests for hibernation particularly important and therefore becoming the focus of many studies [6,7,8]. However, hibernation is a flexible strategy. In warmer climates, such as in Italy, hibernation is reduced by up to two months between January and February [1], and it completely disappears in South Spain [9]. Still, as a secretive nocturnal animal, hedgehogs spend the day in well-hidden day nests [10], which provide security and protection from the elements. Summer nests, in comparison with winter nests, tend to be less robust [3]. Sometimes, during the summer days, animals simply rest under the cover of vegetation. When asleep by day, hedgehogs do not roll up tightly but sleep on their side, with their muscles and spines relaxed [3]. In warm weather, they may even stretch out. Given the vulnerability that such a posture offers, a safe choice of day-nest location becomes crucial, not only during wintertime but all year around.

The European hedgehog is a species of conservation concern in Europe due to their reported population declines [11,12]. One of the main causes for that decline includes habitat loss due to land use changes like agricultural intensification and urbanisation. Reducing and removing a key element, such as hedgerows to increase field sizes, has become an important threat to hedgehogs [13]. Therefore, conservation strategies focusing efforts on hedgerow management have proven to be beneficial for biodiversity in agricultural-dominated landscapes [14]. In line with this idea, we should not forget the constant growth of cities. There has also been rapid urban expansion accompanied by an increasing tendency for mammals to colonise urban settlements [5]. Urban environments and metropolitan areas will then play an increasingly important role in conserving biodiversity at the local and regional levels. Hedgehogs are classified as urban adaptors [15,16], and recent studies have highlighted that hedgehogs are more likely to be found in towns and cities than in rural areas [2,17,18,19] and have higher densities in urban areas [5,20]. Given this scenario, it is important to find what drives nest habitat use by hedgehogs in urban environments. The availability of safe nest locations may be important for their survival in urban areas and for maintaining stable populations.

The aim of this study was to characterise for the first time the key elements of habitat use for nesting in urban populations of European hedgehogs at the southern range of this species distribution. In order to do so, we studied hedgehogs in two areas representative of two types of urban habitats used by this species in the region: The campus of the University of Alicante was used as an example of an intensively gardened park, and Mount Benacantil was used as an urban forest with more natural vegetation. In the present study, we analysed: (1) the number of nests and the frequency of use depending on sex, season and location; (2) the spatial distribution of the nests and their distance to artificial structures; (3) the structures in which the nests were located and the habitat around them; and (4) the nests’ characteristics and nesting materials used.

## 2. Materials and Methods

### 2.1. Study Area

We selected two areas representative of two types of urban habitats used by hedgehogs in the region.

The campus of the University of Alicante is a big, gardened area (80 ha) with an intense and almost continuous human presence, located in the urban area of San Vicente del Raspeig (38°23′05″ N, 0°30′47″ W, Alicante province). Half surrounded by buildings and roads, the landscape is completely flat. The vegetation is green and lush, thanks to its artificial irrigation, with bushes and garden plants that provide several good shelters between areas of pavement and wide-open grass (Figure 1). In contrast, Mount Benacantil (34.8 ha), with less human presence, is within the city of Alicante (38°21′00″ N, 0°28′41″ W). The topography of the mountain is steep (109 m drop), and it perfectly exemplifies the typical dry Mediterranean landscape: scattered vegetation of *Salsola oppositifolia* and *Rhamnus lycioides* as the predominant bushy species, and *Stipa tenacissima*, *Brachypodium retusum* and wide-open spaces of bare soil (Figure 1). As in several public spaces in the cities, we also found several points with cat feeders intentionally delivered by citizens to feed stray cats that were also accessible to the hedgehogs (University campus n = 14; Mount Benacantil n = 5). Those cat feeders are permanently placed in the same locations and are provided daily with new food and water (Figure 1).

### 2.2. Hedgehog Capture and GPS Attachment

Fieldwork was conducted between the 12 December 2016 and 15 March 2017 (winter period) and between 3 June 2017 and 28 September 2017 (summer period) on the campus of the University of Alicante and between 25 March 2017 and 8 October 2017 (spring–summer period) in the urban forest of Mount Benacantil.

Hedgehogs were captured by hand at night with the aid of spotlights. Once captured, all animals were sexed according to the urogenital distance [3,21] and were classified as juvenile, subadult or adult according to their body measurements, weight and the presence of growing spikes [3]. All individuals were marked with a unique colour combination of heat-shrink plastic tubes glued to their dorsal spines [4,22].

From a known population of monitored hedgehogs (university campus n = 25 adults; urban forest n = 22 adults), a sample of 31 hedgehogs (university campus n = 16; urban forest n = 15) weighing >600 g was equipped with tracking devices to study their movements and habitat use. Each device was a pack that included a global positioning system (GPS) and a very high frequency (VHF) transmitter that was attached to a mid-dorsal patch of clipped spines [23,24] using dental composite Protemp^™^ 4 [25,26] (<5% of the hedgehog’s body mass; [27]). The attaching procedure lasted no more than 15 min, and the animals were released at the same point of capture.

To ensure the hedgehogs’ tolerance to the tagging process, a hedgehog was tagged for a trial period of three days. Subsequently, all hedgehogs were monitored for 1 week–10 days, except for one hedgehog that retained the device for 16 days due to some difficulties in relocating it. At the end of the monitoring period, the hedgehogs were caught again thanks to the VHF transmitters that continuously broadcast a signal. They were weighed, and all devices were removed by cutting off the tip of the spines below the dental composite material [26,28].

### 2.3. Logger Sampling Setup and Nest Locations

Hedgehog nests are usually well concealed and very difficult to find. To overcome this problem, the GPS devices were programmed to operate continuously, capturing positions every 3 min during the initial phase of the survey. However, as repeated daytime locations yielded no valuable data, a modified schedule was implemented to conserve the transmitter’s battery life. Accordingly, the GPS devices were set to record locations between two hours before sunset and two hours after sunrise, thus ensuring the location of the nests. As the time of sunrise and sunset varied throughout the study, the GPS devices were programmed week by week in order to adjust them to the desired time. We could only access data and, thus, the location of the nests, once the device was detached from the hedgehog and the data was downloaded.

### 2.4. Nest and Habitat Characterisation

All the nests were visited, and both the habitat around the nest and the nest itself were characterised. We calculated the number of nests used by each hedgehog and the number of times each nest was used during the monitoring days. From the nest coordinates, we calculated the distances (mean, minimum and maximum) between the nests of each hedgehog used on consecutive days. We also calculated the MCPs (minimum convex polygons), including all the nests used by each hedgehog as a measure of their spatial spread, using QGIS 3.26.1.

#### 2.4.1. Nest Characteristics

The nests were examined when possible, but only if a hedgehog was found inside the nest, actively using it at the time of the visit. The nests were classified into four structural categories: well-structured (packed material forming a compact dome), poorly structured (accumulated material, partially covering the hedgehog but without any compact form), unstructured (hedgehogs simply resting under the cover of vegetation, potentially with some contribution of the materials serving as bedding, but without covering them) and burrow (cavities, holes). The dimensions of the well-structured and poorly structured nests were measured: the length, width and height (cm) and nesting materials found in all the nests were described.

For each nest location, we recorded the main structure that hosted the nest, the complementary structure (if present), the minimum height and length (cm) of the main hosting and complementary structures and the distance to the ground of the lower part of the structure at its most distal part to determine the presence or absence of immediate shelter. The principal structure was defined as the main structure hosting the nest of the hedgehog. It was divided into four different categories according to the nature of the structure itself: artificial structure (anything that was man-made, e.g., building, stone wall), shrub-like plants (compact and rigid plants that have woody or similar structures, e.g., bush, branched cactus, branched palm tree), sturdy plants (plants that without having woody or rigid parts form a compact structure, e.g., ivy, pampas grass) and herbaceous plants (plants without woody or rigid parts and with loose structures, e.g., tall grass). The complementary structure was defined as a plant structure that was not part of the main structure but whose presence provided additional shelter. It was divided into two categories: live plants (e.g., loose tall grass, small dense plants) and leaf litter (naturally accumulated and not carried by the hedgehog). A shelter was considered to be present when the hedgehog was fully hidden within the hosting structure immediately upon entering it, either by the main structure alone or through a combination of the main structure and a complementary one. The presence of immediate shelter was recorded when the distance between the ground and the lower part of the hosting structure at its most distal part was less than 10 cm, while an absence was recorded when the gap under the structure was greater than 10 cm.

A value of slope (°) was assigned to each nest using the Digital Slope Model layer—MDP05 (5 m mesh)—downloaded from the CNIG page (National Centre for Geographic Information).

To evaluate the potential effect of the presence of some artificial structures on a nest’s habitat use, the distance to four types of structures was calculated for each nest location using QGIS 3.26.1: distance to the nearest path, distance to the nearest road, distance to the nearest cat feeder and distance to the nearest streetlight. Only accessible paths and roads were taken into account when estimating the distances to those structures, and only the minimum real distance to the feeders was used (e.g., avoiding straight lines crossing buildings or walls).

#### 2.4.2. Habitat around Nests

The habitat’s characteristics were measured for all the nests located, using a 5-metre radius around the central point of each nest, following the methodology used in the SEMICE project (small mammals monitoring in Spain) [29] with a slight adaptation to the elements described. A percentage of cover was assigned to each of the following elements: leaf litter (poorly decomposed plant waste that accumulates on the surface of the soil), bare soil (e.g., sand, pebbles), paved soil (ground covered by several types of pavement, such as bricks or paving stones), rocks > 0.5 m, branches (on the ground), shrub < 1.5 m, shrub > 1.5 m, different species of trees (assigning a percentage category to each species separately), grass-like plants (plants without hardwood, e.g., creeping plants, grass), large infrastructures (buildings) and small infrastructures (small man-made elements).

### 2.5. Statistical Analyses

We analysed whether the number of nests used by an individual, the number of times that each nest was used, their spatial spread (minimum convex polygon, MCP) and the distance to artificial structures (path, roads, cat feeders and streetlights) varied according to sex, locality (university or urban forest) or season (spring–summer or winter). Given that the hedgehogs from the urban forest were not monitored in winter, the effect of the season was analysed using only data from the university campus. When both localities were compared, only data from spring–summer were used. In models that included repeated data from the same hedgehog, generalised mixed models with individual identification as a random effect were used. This included all models at the university because an individual may contribute data for both seasons, while for the urban forest, it included the models for the nest distances to artificial structures and times occupied. These models were fitted with function glmmTMB from the R package glmmTMB [30]. In the models in which each individual contributed just once, GLM was used. In both types of models, the family distribution used was Poisson for the counts (number of nests, number of times occupied) while Gamma (link = “log”) was used for the distances (e.g., distances between consecutive nests, distance to artificial structures) and the MCP.

The number of nests and number of times occupied may be affected by the number of days that an individual was monitored; thus, to take into account this effect, the number of monitoring days was included as a covariate in the models for these variables. Only hedgehogs with more than 1 nest were used in the models for the mean distances between consecutive nests, and only the hedgehogs with more than 2 nests were used in the MCP models. Since the number of recorded nests could affect the MCP values and the distance between them, the number of nests was included in these models as a covariate. First, the models that included the interaction of sex and season (university data) or sex and place (comparison of study sites), including the mentioned covariates, if necessary, were fitted and then were simplified, removing variables using the AIC as a criterion [30].

We used factorial analysis of mixed data (FAMD) to investigate the gradients of the environmental variables related to nest placement. The FAMD allows continuous and categorical data to be integrated into the same analysis. Thus, it allowed us to investigate the association between the categorical variables describing the location of the nest (principal structure, complementary structure and shelter) and the continuous variables relating to the cover of soil around the nests. We used the R package FactoMineR [31] to perform the FAMD. The dimdesc function was used to identify the variables with a stronger relation to each dimension using correlation in the case of continuous variables, and the significance and R^2^ of an ANOVA were used when testing for the differences of coordinates between levels of each categorical variable [31].

The nest coordinates in the selected dimensions, resulting from this analysis, were used as dependent variables in linear mixed models (glmmTMB package) to test the fixed effects of locality, season and hedgehog sex on nest location and environment. In this analysis, the hedgehog’s identity was used as a random effect. The significance level for statistical analyses was set at *p* = 0.05.

## 3. Results

### 3.1. Seasonal and Sex Differences within Localities

A total of 130 nests (59 at the university and 71 in the urban forest) belonging to 31 hedgehogs were found. The results from the university campus showed that the number of nests per hedgehog and the distance between the consecutive nests did not vary between seasons or sexes (Table 1, Appendix A). The values of the MCPs were compared between seasons only using data from males, as there were no females in summer with more than two nests found. The results showed only a trend for the nests of males being distributed in a wider area in summer (Figure 2).

The interaction between sex and season was significant for the number of times a nest was occupied; thus, females used the same nest more times during summer and changed nests more often in winter, while the season had no effect on males’ nest occupations. The distance from a nest to the nearest cat feeder did not vary between seasons or sexes. On the contrary, males tended to locate their nests farther from paths and streetlights but closer to roads than females (Table 1, Figure 2, Appendix A).

All nests studied in the urban forest were found during the spring–summer period, limiting the analysis of the seasonal effects in this location. With respect to the sex’s effect, the distances between consecutive nests did not differ between the sexes, but the spread of the nests (MCP) was found to be greater for males (*p* = 0.031).

### 3.2. Differences between Localities in Spring–Summer

The spatial spread of nests was similar in both study areas; nevertheless, the hedgehogs had, on average, fewer nests at the university campus. However, the interaction between sex and locality approached significance (Table 2), suggesting that females have a lower number of nests than males at the university but not in the urban forest. In both localities, males occupied the same nest less frequently than females, and there was a trend for the nests at the university being used more times than the nests in the urban forest (Table 2, Appendix A).

Most of the distances to artificial structures differed between the study areas and, in some cases, between the sexes. The nests were located closer to streetlights at the university, and in both areas, the male nests were farther from them. There was a trend for the nests of both sexes to be closer to cat feeders in the urban forest. The interaction between sex and locality was significant for the distances to paths and roads, indicating different responses by sexes to these structures in each place. In the university, the females placed nests, on average, farther from the roads than the males, but contrarily in the urban forest, the nests of females were closer to roads (Figure 3, Appendix A). In the case of distance to paths, the pattern is the opposite, with male nests being farther than female nests from the paths at the university and closer to paths in the urban forest.

The slope on which the nests were located varied greatly between zones due to the different topographies of the study areas (flat at the university and with a steep slope in the urban forest). The average slope for the nesting sites in the urban forest was 25°, reaching maximum values of 36.5° for female and 41.5° for male nests.

### 3.3. Nest Characteristics

We examined a total of 15 nests in both study areas that were actively used by a hedgehog at the moment of its location (Table 3, Appendix A). A total of 46.7% of the nests examined were classified as unstructured since the hedgehogs were found simply resting under the vegetation with no or only some material (presumably carried by the hedgehog) around but without any organisation and without covering the hedgehog. The unstructured nests were found both in summer and winter (Table 3). On the other hand, 26.7% of the nests were considered to be well-structured nests, with a significant amount of nesting material accumulated around the hedgehogs and covering them, forming a compact dome. Poorly structured nests (nests with some material partially covering the hedgehogs but without a robust structure) and burrows represented 13.3% (of each) of the nests examined. Nesting materials, presumably carried by the hedgehogs to the nest, mainly included the dry leaves of tree and/or shrub species present in each area, such as *Pittosporum* or bamboo, and dry grass. However, 27.7% of the nests also included man-made objects, such as torn plastic bags, pieces of fabric, chocolate wrappers and even an umbrella.

### 3.4. Nest Location and Habitat around Nest

Almost all the nests in the urban forest were located under shrub-like plants, mainly *Salsola oppositifolia* (Table 4, Appendix A). Nest locations at the university were more varied, and only about half of the nests were located at shrub-like plants, and the number of plant species used was also more diverse (Table 4, Appendix A). The diversity index of the nest locations sites was significantly higher for the university (Hutcheson test, t = 8.67, d.f. = 118.7, *p* < 0.001). In both study areas, the diversity index of the nesting sites used by males appears to be higher than by females; however, this difference is only significant for the university (Hutcheson test, t = 2.34, d.f. = 23.8, *p* = 0.028).

The FAMD identified seven dimensions with eigenvalues > 1. We selected to interpret the first five dimensions on the basis of presenting strong correlations with more than one continuous variable that, after inspection of the scatter plots (Appendix A), were not caused by some extreme values. Table 5 shows the indicators of the relation of each variable and the dimensions, and Figure 4 shows the ordination of nests in the first four dimensions. The first dimension is related mainly to the cover of large artificial structures and secondarily to the cover of leaf litter and separates the nests located at artificial structures with no shelter (most positive values), mainly found at the university, from the nests located at shrub-like plants (negative values). Only 18.2% of the artificial structures provided immediate shelter compared to 95.8% of the other nesting sites. The second dimension is positively related to the cover of grass-like plants and negatively related to shrub cover and separates the nests located at sturdy or herbaceous plants with live plants as complementary structures (positive values) from the nests located at shrub-like plants with complementary structures made of leaf litter (negative values). The third dimension presents a positive correlation with the cover of paved soil and shrubs < 1.5 m and a negative correlation with large ones (>1.5 m); however, there is no clear separation of the types of nest placements along this gradient. The fourth dimension is positively correlated to the cover of trees and leaf litter and separates the nests located at herbaceous plants with a complementary structure of leaf litter from the nests located at shrub-like plants with a complementary structure of live plants. Finally, the fifth dimension is positively correlated to the cover of bare soil and small artificial structures and separates the nests located at herbaceous plants (positive values) from the nests located at sturdy plants (negative values).

According to the linear mixed models (Table 6), nest location differed between the university campus and urban forest in the gradients defined by the first four dimensions and differed between males and females in Dimensions 2 and 3 (Figure 4). Nests in the urban forest were rarely located in artificial structures, while about 17% of nests at the university were located in them. Nests at the university have, on average, more positive coordinates in dimension two, associated with a higher cover of grass-like plants and nest placement on sturdy or herbaceous plants—types of location where nests were not found or rarely found, respectively, at the urban forest. The differences in Dimensions 3 and 4 were associated with nests at the university having around a larger cover of paved soil, leaf litter and trees than the nests at the urban forest. Interestingly, there were consistent differences between the sexes at both study areas, with male nests having, on average, lower coordinates in Dimensions 2 and 3 than females. According to the description of the dimensions, this may be interpreted as nests of males being located more often than female nests in areas with more cover of large shrubs and fewer grass-like plants. In the case of the university, this is also associated with the nests of males being more frequently located at shrub-like plants than females, who place their nests more often at sturdy plants.

## 4. Discussion

Our results have revealed differences between male and female hedgehogs regarding their various aspects of nest use, spatial spread and location in relation to habitat’s characteristics. These differences are likely related to the promiscuous mating system of the species [3,32,33] and the fact that males’ home ranges are between two to five times larger than females [1,7,9,10,24]. We also found some differences between the urban habitats studied, which are likely related to differences in the topography and types of habitats available at each locality.

The distances between consecutive nests did not differ between sexes; however, in the urban forest, we estimated that the males presented a larger nest spread, as measured by the minimum convex polygon (MCP) of the nests, compared to the females. We could not test the sex effect at the university campus due to limited data; however, we observed a trend for the male nest MCP being larger in spring–summer compared to winter. Both of these patterns align with expectations due to the larger home range of males and their active search for mating partners [24]. The combination of both results together shows that the exploratory behaviour of males may be staggered. Males do not change areas drastically from one night to the next (the distance between consecutive nests is similar to that of females); however, it is the cumulative effect over several nights that contributes to larger home range areas for males. Furthermore, we discovered that females exhibited a higher frequency of nest reuse compared to males, with this difference being more pronounced during the summer period. In general, females’ activity is restricted to a smaller area, making it feasible to return to the same nest daily and perhaps this also being more cost-effective in energy terms [3,34]. This limited home range is accentuated, especially during the breeding season, when a reduction in home ranges has been observed in female hedgehogs [24,35], as in other small mammals [36,37].

Hedgehogs spend the entire day sleeping in their resting nests [3]. Good protection against predators and threats during this vulnerable time is necessary, especially in urban environments, where the daily movement of people and dogs is frequent. Therefore, it is likely that females’ selectivity of nesting locations extends to their choice of cover or structure for nest placement, as well as the surrounding environment. None of the females tagged during the spring–summer period were pregnant or had hoglets, but it is likely that nest selectivity extends beyond the breeding season, possibly even throughout most of the year, as hedgehogs in the Mediterranean region may also breed in autumn and winter (pers. obs.).

The diversity of plants or structures used by females for nest placement was found to be lower compared to males in both study areas. However, this difference reached statistical significance only at the university campus, where it was more pronounced. At the university campus, we identified a total of 23 distinct types of nest locations, while in the urban forest, we found only 12 types. This suggests that the presence of gardens and artificial structures on the campus provides a wider array of potential nesting places [35]. Hedgehogs utilised six different types of artificial structures at the university campus, whereas they only utilised one type (a hole in a wall) in the urban forest. We lack estimates regarding the abundance of potential nesting sites; however, considering the university campus has a larger surface of built areas, it is expected that the availability of artificial structures would be greater there. Consequently, we cannot determine whether the higher frequency of nesting in artificial structures at the university campus indicates a preference for them or simply results from their greater availability.

The FAMD ordination of nesting sites and their environment clearly revealed a distinction between the two urban study areas, which was statistically significant in the first four dimensions, likely indicating variations in nesting sites’ availability and habitats. The first dimension primarily separates nesting in artificial structures lacking shelter at their entrance from nesting under plants, which typically provide shelter. Nesting in artificial structures is primarily found at the university, although nesting under plants remains predominant there. Interestingly, this analysis revealed differences between the sexes in Dimensions 2 and 3, providing evidence of variations in the average nest selection environments between males and females. Dimension 2 depicts a gradient ranging from nesting in sturdy plants with a complementary structure and ample grass-like plant cover in the surrounding area (positive values of the dimension) to nesting beneath shrub-like plants in an area dominated by large shrubs (negative values of the dimension). Interestingly, and consistently across both locations, females nested more often in the former and males in the latter. Dimension 3 is exclusively correlated with the surrounding habitats of the nests and represents a gradient ranging from areas characterised by paved soil and small shrubs (positive values) to areas dominated by large shrubs (negative values). In this gradient, there is also a consistent difference between the sexes across the two locations, with the majority of female nests situated in areas with a higher abundance of small shrubs and, in the case of the university, more paved soil.

Within the group of sturdy plants where the females established their nests at the university, ivy dominated, followed by pampas grass. Both plant species, unlike many shrubs, are difficult-to-access structures, meaning that they provide highly concealed protection for hedgehogs, with large surfaces around them (e.g., ivy) and/or thick walls of plant tissue (e.g., pampas grass). These types of plants were absent in the urban forest, where all female nests were found within shrub-like plants, primarily *Salsola oppositifolia*, which was the dominant shrub species in that area. This raises the question of why female nests were more frequently located in sturdy plants at the university campus, while in the urban forest, they were able to utilise mostly shrubs. The answer is likely related to the variability in shape and density of the shrubs, as not all of them would be equally suitable for female nests. All plants used by females at the university and included in the category of shrub-like plants, such as *Pittosporum*, *Lavandula* or *Cupressus*, are quite compact. In contrast, as well as these same species, the males made nests in other shrub-like plants (e.g., *Acacia*, *agapanthus*, *Nerium oleander*) that were never used by the females. Those bushes, unlike others, have less dense branching structures and seem to provide a lower level of protection that may not be suitable for females.

The presence or absence of an immediate shelter, which allows hedgehogs to quickly hide when entering the structure that includes the nest, seems to be important, given that the great majority of used sites (89.2%) offer this type of shelter. Among the shrub-like plants, immediate shelter was present and directly provided by the principal structure in 96% of the cases. Of the remaining 4%, in 26.3% of the cases, shelter was provided by the presence of the secondary structure, which could contribute to increasing its suitability. The nature of the secondary structures varied greatly between localities. In the urban forest, for 93.3% of nests, the secondary structure was composed of live plants, mainly *Brachypodium retusum*, while at the university campus, it was predominantly composed of leaf litter (66.7% of the cases). Therefore, at the university campus, leaf litter seems to contribute to the increased suitability of some shrub-like plants besides being a component of the structure and inner lining of the nest. In addition, leaf litter is known to be an important source of food, as it provides a habitat for their natural prey, such as snails, slugs and insects [3]. Therefore, removing leaf litter from gardens and managed urban parks is likely to contribute to a decreased habitat quality for hedgehogs and is not recommended [14].

The artificial structures were the only ones that lacked immediate shelter in most cases, with only 18.2% of the used sites having shelter. These structures also did not have a complementary structure that could contribute to hiding the hedgehog. As a consequence, the hedgehog was not immediately protected by the structure when entering and, thus, had to go deeper. We recorded only two cases of artificial structures used by females: a nest box specifically designed for hedgehogs, and therefore likely to be more suitable than the average artificial structures, and an office trailer, whose size and shape could have contributed to being selected by females. But overall, artificial structures were used mainly by males, which supports the idea that males are less selective of nesting sites.

The type of sites used for the nest and the characteristics of its surrounding environment are expected to be related, as some types of structures are more likely to be found in particular habitats. An example of this effect may be the tendency of females to place nests in sites with greater cover of paved soil than males, especially at the university. This could be a consequence of the females selecting (less often than males) shrub-like plants for their nests, as there is a negative relation between the cover of large shrubs and paved soil. On the other hand, females showed a tendency to choose sites with greater grass-like plant cover around. This result may originate from the selection of nesting sites with greater availability of food items nearby since the amount of time spent by females searching for natural prey in lawns has been shown to be greater than that spent by males in urban areas where cat feeders are available [9]; likewise, female nests were located closer to the streetlights compared to male nests. Although most studies report negative behavioural consequences of artificial light at night (ALAN) in mammals [38], some species, like hedgehogs, are able to exploit the foraging opportunities created by lighting, such as the accumulation of insects around streetlights [39].

The contrasting topography of the university campus (flat) and the steep urban forest located on a mountain may affect the cost of movement [40] and help to explain some differences in nesting ecology between the study areas. At the university, hedgehogs used fewer nests and used each nest more frequently than in the urban forest. Due to the university campus’s flat landscape, hedgehogs can move between nests at a lower cost and need fewer nests to cover their home range compared to the urban forest. Similarly, proximity to stable food sources, like cat feeders [41], might be more important in areas where the costs of movement are larger. Nests of both sexes were closer to the five available cat feeders in the urban forest but were more distant at the university campus with fourteen feeders.

Differences in the management of the urban parks studied may have also contributed to increasing the dependency on cat feeders in the urban forest. The artificial irrigation employed at the university could enhance the availability of invertebrates, whereas, in the dry and arid urban forest, hedgehogs may rely more on supplementary food from cat feeders. The different distributions of artificial structures between the study areas may also help to explain some of the contrasting patterns found. Roads at the university campus are close to the periphery as it is designed as a pedestrian area, while in the urban forest, there is a road running roughly through the centre. The males placed nests closer to roads at the university campus but farther away in the urban forest. This sex and locality interaction could be attributed to the distinct road distribution. The males’ tendency for longer exploratory movements and larger home ranges [1,7,9,10,24] would make it more likely that some male nests will be located near the periphery, where roads are present at the university but farther from the central road in the urban forest. Hence, this pattern is more likely a result of the sex-related differences in behaviour rather than a deliberate choice for habitats near roads.

Finally, our results showed that the nests described here are looser and smaller than those described at northern latitudes [3,8,35]. Many authors have stated that there is a certain plasticity in the construction of the nests [34]. Hibernacula in the northern latitudes must be well-sited, robust and carefully constructed to protect torpid animals over extended periods [7,35]. Captive animals did not build a nest when the ambient temperature exceeded 16 °C [42], resulting in relatively less substantial summer nests compared to winter nests. The mild winter temperatures in the Mediterranean region may have led to a less robust nest construction compared to the hibernacula found in the northern regions [35]. Nests from all seasons in our study areas were relatively flimsy, constructed merely of grass and dry leaves, and poorly compacted, as occurs in other mild climates, such as in New Zealand [43,44], where the species has been introduced. Adequate construction materials for nest-building limit hedgehog distribution in higher altitudes, often linked to deciduous forest distribution [3]. In urban areas, planted deciduous trees provide suitable materials, as seen in our results, with 40% of the nests composed of dry leaves and 66.7% with dry grass. The hedgehogs also used other artificial resources as bedding materials, such as plastics and chocolate wrappers [35], found in 26.7% of the nests. Similar results to those found by Schoenfeld and Yom-Tov with *Erinaceus concolor* in Israel [45] reaffirms the generalist and opportunistic nature of these species.

## 5. Conclusions

Our study has shown differences in the nest distribution and habitat use between male and female hedgehogs in urban habitats. Females showed a preference for well-concealed structures with the presence of immediate shelter, while males were able to use a greater variety of nesting sites and tended to locate their nests further apart than females. Females tend to reuse the same nest more often than males. These results suggest that females are more selective than males when choosing the structures hosting their nests. There are also differences between the sexes in the distance of nests to some artificial elements present in the urban habitat that may act as barriers, sources of perturbations or, contrarily, attraction points. Females nested closer to paths and streetlights at the university, while males were closer to roads. Opposite results were found in the urban forest, where a road ran through the centre and where female nests were closer to roads than those of males, which is evidence that the distribution of elements in urban areas is important. Different topographies and garden management also affected the dependence of hedgehogs on supplementary food, with the nests closer to the cat feeders in the urban forest.

Nests in both study areas were found to be relatively flimsy, consisting merely of grass and dry leaves, with poor compaction, and significant changes in nest characteristics were not detected across the seasons. This contrasts with the results from northern latitudes, where winter nests are thicker and far better constructed than spring–summer nests, pointing out that in southern latitudes, the selection of good hosting structures seems more important than the nest construction itself.

Differences in nesting ecology between the two urban parks observed are likely to be related to their different topography and habitat characteristics and availability, highlighting the importance of studying hedgehogs’ nest site selections at the microhabitat level. However, our study has some limitations because, despite having analysed more than a hundred nests, they were built by just 31 hedgehogs, which were monitored for a limited period of time. Consequently, the sample sizes of some combinations of sex, season and study site were limited, and we were unable to fully explore the effects of these factors. Future studies with larger sample sizes and in other habitats are needed to confirm or not the generality and strength of the patterns we have found. This type of study would contribute to a better understanding of hedgehogs’ nesting requirements, fostering the design of more effective conservation strategies for hedgehogs and the appropriate management of green urban areas for this species and other wildlife.

Currently, based on our results, we can highlight certain practices that would enhance the suitability of garden habitats for hedgehogs’ nests. These include: (1) using plants with a dense structure, where branches begin close to the ground to provide good shelter for the hedgehogs; (2) maintaining lawn areas as naturally as possible [46]; and (3) preserving leaf litter in gardens. Furthermore, it is important to consider the possibility of hedgehog nests when conducting maintenance work in gardens, as hedgehogs may utilise unexpected temporary or movable structures for nesting. Finally, our research revealed that hedgehogs often selected nesting sites near roads, which are a significant cause of mortality for the species [47]. Therefore, where suitable nesting sites are particularly close to roads, corrective measures are encouraged to prevent road casualties.

## Figures and Tables

**Figure 1 animals-13-02453-f001:**
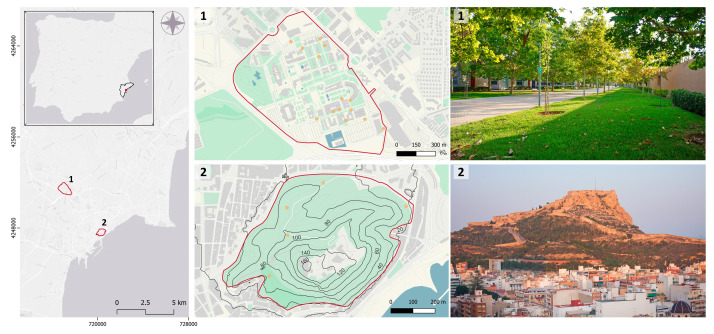
Locations of the two urban areas studied in the province of Alicante (left map) and maps of their habitat composition, along with representative photos: (1) Campus of the University of Alicante; (2) urban forest (Benacantil). Key habitat elements are mapped with different colours: green areas (green), buildings (grey), water (blue), cat feeders (orange dots), paths (dotted grey lines), roads (white lines) and contour lines (black lines).

**Figure 2 animals-13-02453-f002:**
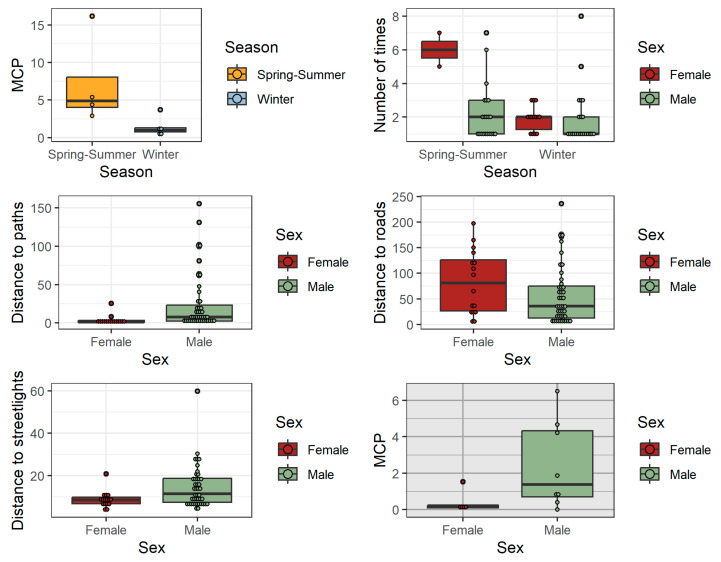
Distribution of hedgehog nest variables measured at the university campus (white background plots) or urban forest (grey background plot). Only variables that showed an effect of season or sex, with *p* < 0.1 as determined by GLM or GLMM (in Table 1 for the university nests), are presented. The MCP (minimum convex polygon) analysis for university data included only males. Typical boxplot structure (median, lower and upper quartiles, and whiskers calculated as 1.5 times the interquartile range) is shown together with the actual values of each variable (small circles).

**Figure 3 animals-13-02453-f003:**
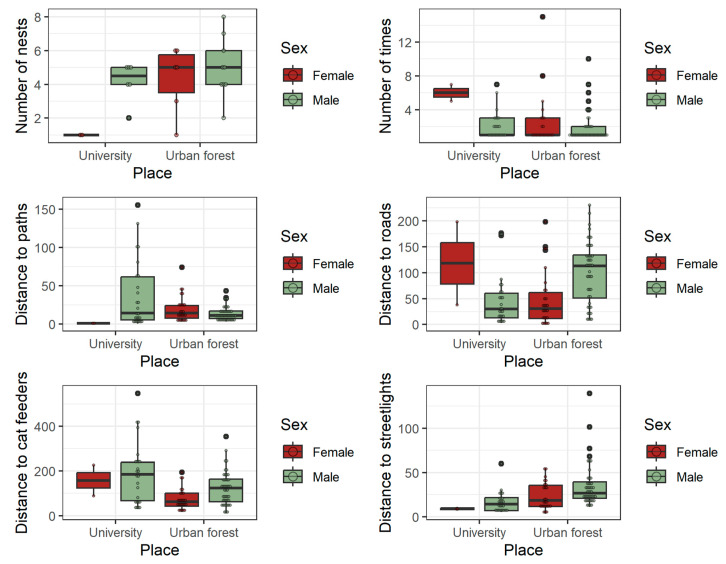
Distribution of hedgehog nest variables measured at the university campus and the urban forest during the spring–summer period. Only variables that showed an effect of season or sex with *p* < 0.1, as determined by GLM or GLMM (see Table 2), are presented. Typical boxplot structure (median, lower and upper quartiles, and whiskers calculated as 1.5 times the interquartile range) is shown together with the actual values of each variable (small circles).

**Figure 4 animals-13-02453-f004:**
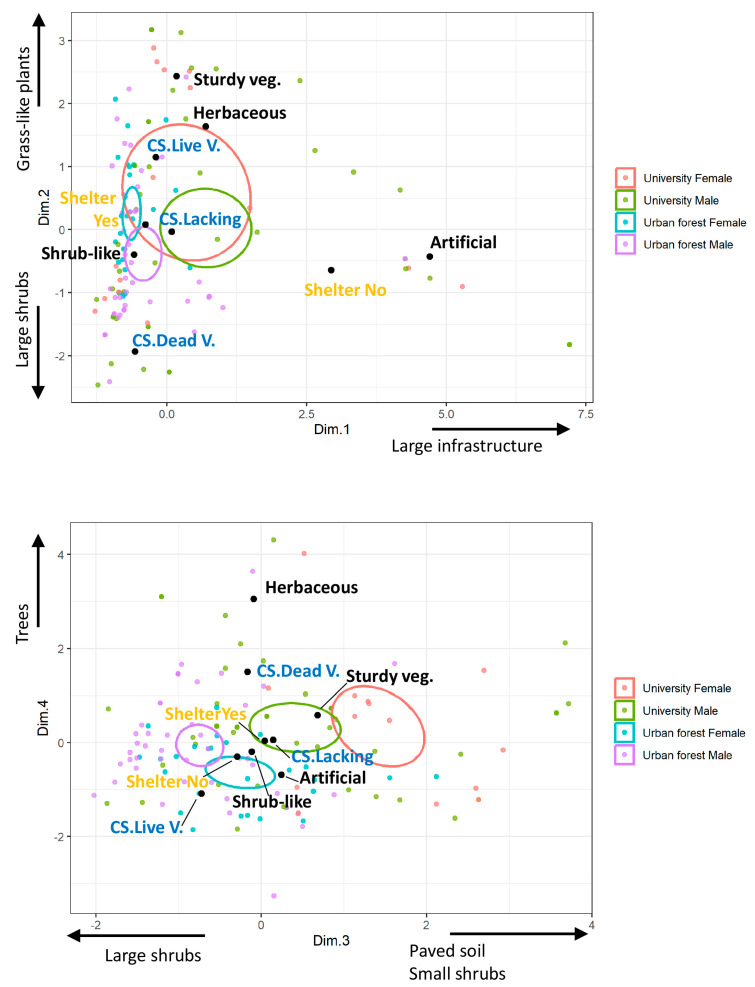
Ordination of hedgehog nests in the first four dimensions resulting from factorial analysis of mixed data (FAMD). Nests are represented with points coloured according to the study area and sex. Categorical variable levels are indicated by black dots, representing the average scores of nests within each level. The colour of the level’s text identifies categorical variables: black (main structure hosting the nest), blue (C.S. complementary structure, whether dead or live vegetation or lacking), and orange (presence or absence of shelter). The 95% confidence ellipses are shown with colours corresponding to the study area and sex. The arrows identify variables with stronger correlations with each dimension (refer to Table 5 and Appendix A for more information).

**Table 1 animals-13-02453-t001:** Models (GLMM) for variables relating to the set of nests used by a hedgehog and variables relating to each nest use and distance to artificial habitat features at the University of Alicante campus. Positive or negative signs of variables retained in the final model are included, as well as their *p*-values. n.r.: variables not retained in the final model. Cells for variables not used in a model appear empty. The variables number of monitoring days (M. Days) and number of nests found for each individual (No. nests) are included in some models to control for their potential effects on some dependent variables. Season: reference level, winter. Sex: reference level, female.

Variable	Season	*p*	Sex	*p*	Period × Sex	*p*	M. Days	*p*	No. Nests	*p*
Number of nests		n.r.		n.r.		n.r.	+	0.4603		
Distance consecutive nests ^1^		n.r.				n.r.			+	0.3854
MCP nests ^1^	+	0.0898							+	0.6996
Times occupied	+	0.0034	−	0.5931	−	0.0021	+	0.0072		
Distance to path		n.r.	+	0.0014		n.r.				
Distance to road		n.r.	−	0.099		n.r.				
Distance to cat feeder		n.r.		n.r.		n.r.				
Distance to streetlight		n.r.	+	0.0027		n.r.				

^1^ Model for MCP and distance of consecutive nest fitted to only male data.

**Table 2 animals-13-02453-t002:** Models for variables relating to the set of nests used by a hedgehog (GLM) and variables relating to each nest use and distance to artificial habitat features (GLMM) at the University of Alicante campus and the urban forest. Positive or negative signs of variables retained in the final model are included, as well as their *p*-values. n.r.: variables not retained in the final model. Cells for variables not used in a model appear empty. The variables number of monitoring days (M. Days) and number of nests found for each individual (No. nests) are included in some models to control for their potential effect on some dependent variables. Place: reference level, urban forest. Sex: reference level, female.

Variable	Place	*p*	Sex	*p*	Place × Sex	*p*	M. Days	*p*	No. Nests	*p*
GLM										
Number of nests	−	0.0346	+	0.6223	+	0.0808	+	0.4872		
Distance consecutive nests ^1^		n.r.							+	0.2882
MCP nests ^1^	+	0.1190							+	0.2430
GLMM										
Times occupied	+	0.0682	−	0.0283		n.r.	+	0.0017		
Distance to path	−	0.0000		0.3990	+	0.0000				
Distance to road		0.1754	+	0.0070	−	0.0131				
Distance to cat feeder	+	0.0443		n.r.		n.r.				
Distance to streetlight	−	0.0000	+	0.0129		n.r.				

^1^ Model for MCP and distance of consecutive nests fitted to only male data.

**Table 3 animals-13-02453-t003:** Characteristics of the 15 nests analysed in detail that were actively used by hedgehogs at the time of the visit. Study areas: urban forest (Benacantil); university campus (UA).

Nest ID	Age & Sex	Month	Study Area	Hosting Structure	Type of Nest	Nest Dimensions (cm)	Nest Material
*T6*	Juv F	May	Benacantil	Shrub-like plant	Unstructured		Grass
*K5*	Ad F	June	Benacantil	Shrub-like plant	Well-structured	42 × 35 × 31	Grass, dry leaves and pieces of plastic bags
*Pi6*	Ad F	June	Benacantil	Shrub-like plant	Unstructured		Grass
*Gu5*	Ad M	June	Benacantil	Shrub-like plant	Unstructured		Grass and plastics
*FH4*	Ad M	June	Benacantil	Shrub-like plant	Poorly structured	48 × 44 × 31	Grass
*Lu1*	Ad F	August	Benacantil	Shrub-like plant	Poorly structured	50 × 50 × 70	Grass, dry leaves, plastic bags, fabric and an umbrella
*Y2*	Juv M	September	Benacantil	Shrub-like plant	Unstructured		Grass, dry leaves and pieces of plastic bags
*Ñ2*	Ad F	October	Benacantil	Shrub-like plant	Well-structured	30 × 50 × 60	Grass
*L5*	Ad F	October	Benacantil	Shrub-like plant	Unstructured		Grass
*S4*	Ad M	October	Benacantil	Herbaceous plant	Burrow		Grass stump
*B2*	Ad M	January	UA	Shrub-like plant	Well-structured	60 × 39 × 30	Dry leaves Bamboo
*C1*	Ad F	February	UA	Shrub-like plant	Well-structured	53 × 35 × 16	Dry leaves *Pittosporum* sp.
*Rl5*	Ad M	June	UA	Herbaceous plant	Unstructured		Grass
*H3(v)*	Ad M	June	UA	Sturdy plant	Burrow		Palm tree stump
*CR4*	Juv M	December	UA	Shrub-like plant	Unstructured		Dry leaves, plastic bags and chocolate wrapper

**Table 4 animals-13-02453-t004:** Percentage of nests located in different main structures and Shannon index of diversity (H’) of structures used by each sex at each study area. The categories in which the nesting sites were grouped, along with their respective percentages, are shown in bold font.

	University		Urban Forest	
	Female	Male	Female	Male
**Shrub-like plants**	**37.50**	**55.81**	**100.00**	**95.56**
*Acacia dealbata*	0.00	2.33	-	-
*Agapanthus* sp.	0.00	2.33	-	-
*Agave* sp.	0.00	4.65	-	-
*Bamboo*	6.25	11.63	-	-
*Cupressus sempervirens*	6.25	6.98	-	-
*Genista* sp.	-	-	3.85	0.00
*Lavandula* sp.	6.25	2.33	-	-
*Lonicera* sp.	-	-	0.00	2.22
*Mesembryanthemum* sp.	6.25	2.33	-	-
*Nerium oleander*	0.00	4.65	-	-
*Origanum* sp.	6.25	0.00	-	-
*Osyris quadripartita*	-	-	0.00	6.67
*Phagnalon saxatile*	-	-	3.85	0.00
*Phoenix* sp.	-	-	0.00	2.22
*Pistacia lentiscus*	0.00	4.65	0.00	4.44
*Pittosporum* sp.	6.25	13.95	-	-
*Rhamnus lycioides*	-	-	0.00	2.22
*Rosmarinus officinalis*	-	-	3.85	0.00
*Salsola oppositifolia*	-	-	84.62	77.78
*Tetraclinis* sp.	-	-	3.85	0.00
**Sturdy plants**	**43.75**	**16.28**	**-**	**-**
*Cortaderia selloana*	6.25	4.65	-	-
Palm tree stump	0.00	2.33	-	-
*Hedera helix*	37.50	9.30	-	-
**Herbaceous plants**	**6.25**	**9.30**	**0.00**	**2.22**
*Graminea* sp.	6.25	4.65	0.00	2.22
*Phragmites australis*	0.00	2.33	-	-
*Stipa tenacissima*	0.00	2.33	-	-
**Artificial structure**	**12.50**	**18.60**	**0.00**	**2.22**
Wooden box	6.25	0.00	-	-
Office trailer	6.25	0.00	-	-
Building	0.00	11.63	-	-
Stone wall	0.00	2.33	0.00	2.22
Metal ramp	0.00	2.33	-	-
Road tunnel	0.00	2.33	-	-
No. of nests	16	43	26	45
No. of hedgehogs	6	10	6	9
Diversity H’	2.10	2.77	0.64	0.94

**Table 5 animals-13-02453-t005:** Description of the first seven dimensions resulting from a factorial analysis of mixed data (FAMD) of hedgehog nests from both study areas. For continuous variables, the correlation with scores in each dimension is shown (highlighted in bold font are correlations > 0.45, which are displayed graphically in Appendix A). For categorical variables, the table presents the R^2^ resulting from an ANOVA that compares the average scores of each variable’s levels within each dimension (only results for significant ANOVAs are shown).

	Dim.1	Dim.2	Dim.3	Dim.4	Dim.5	Dim.6	Dim.7
eigenvalue	2.56	1.98	1.66	1.56	1.45	1.26	1.10
Cumulative % of variance	16.02	28.37	38.75	48.51	57.55	65.41	72.27
Continuous variables (habitat around the nest)							
Leaf litter	**0.472**	−0.250	−0.009	**0.484**	−0.008	0.362	0.189
Grass-like plants	−0.126	**0.859**	−0.009	0.106	−0.270	0.148	−0.037
Bare soil	−0.020	0.191	−0.273	−0.086	**0.696**	0.083	−0.040
Paved soil	−0.091	−0.214	**0.777**	0.088	−0.056	−0.089	−0.151
Rocks	−0.073	−0.111	−0.002	−0.298	0.021	−0.017	0.760
Branches	0.097	0.183	−0.057	0.027	0.240	−0.264	−0.123
Small shrubs	−0.309	−0.192	**0.590**	−0.401	0.286	0.147	0.002
Large shrubs	−0.318	**−0.489**	**−0.676**	0.129	−0.269	−0.254	−0.081
Trees	−0.173	−0.178	0.265	**0.636**	0.081	0.059	−0.231
Large infrastructure	**0.856**	−0.067	0.061	−0.168	−0.241	0.157	0.044
Small infrastructure	0.232	0.256	−0.045	0.213	**0.569**	−0.403	−0.159
Categorical variables (structure containing the nest)							
Main structure	0.843	0.453		0.343	0.279	0.204	0.183
Complementary structure		0.225		0.206		0.522	0.153
Shelter	0.441					0.087	

**Table 6 animals-13-02453-t006:** Results of the linear mixed models fitted to the nest scores in each of the first five FAMD dimensions. Each model is an additive model with study area (place: reference level urban forest), sex (reference level: female) and season (reference level: spring–summer) as fixed factors and hedgehog identity as random factors. Only significant effects and their sign are shown. UA: Alicante University campus. n.s.: *p* > 0.1.

Dimension	Place	*p*	Sex	*p*	Season	*p*
1	+ UA	0.0023		n.s.		n.s.
2	+ UA	0.0395	− Male	0.0118		n.s.
3	+ UA	0.0000	− Male	0.0013		n.s.
4	+ UA	0.0028		n.s.		n.s.
5		n.s.		n.s.		n.s.

## Data Availability

Data are available upon reasonable request to the authors.

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
