# Peer review of "Nesting Ecology of European Hedgehogs (Erinaceus europaeus) in Urban Areas in Southeast Spain: Nest Habitat Use and Characteristics"

_animals, 2023, doi:10.3390/ani13152453_

Round 1
Reviewer 1 Report
See attached file

See attached file
Author Response
Response to Reviewer 1 Comments
The paper is generally well produced and presents some interesting results on hedgehog nests in Spain. This is a poorly researched area and this work adds usefully to the field
Comments:
1) The following statements were made:
- Lines 9-11 However, little attention has been paid to them in the literature, beyond the study of the hibernacula (winter nests) in northern latitudes,
- Lines 22-23 However, little attention has been paid to them apart from the study of hibernacula at northern latitudes.
This suggests that only hibernation nests have been studies but at least one study of summer nests is in the reference list [34] and referred to later in the text.
AR: The authors agree with the reviewer and apologize for that. The simple summary and the abstract underwent many modifications to fit in 200 words and that sentence was oversimplified. We have modified the lines in both (Simple summary and Abstract) so that it does not give the impression that only winter nests have been studied in the past.
2) I do not understand the term ‘paved soil’. It needs clarification.
AR: Done. We have clarified it in the text.
3) I believe that the special issue is using UK English, so it would be appropriate to change the American English spellings to UK English: ‘metre’, ‘centre’, ‘behavioural’.
AR: Done. We have changed the whole text to UK English, so words like ‘metre’, ‘centre’, ‘behavioural’ and ‘analyse’ are now used.
4) Line 105 would be the best moment to clarify that the food in the cat feeders was accessible to the hedgehogs. Otherwise, the reader has to wait until near the end of the paper to find that out.
AR: Done. We have added a sentence in the text to make it clearer.
5) Line 249 onwards: The Results The results generally only report the test results (which are tabulated) but the actual figures are not reported in the text, or in a table. So I read (line 251) that “The results of the university campus showed that the number of nests per hedgehog and the distance between consecutive nests did not vary between seasons nor sexes (Table 1)” but I am not given any figures. Hedgehogs had fewer nests at the university than in the forest (line 300) but I want to know how many? To get some idea I must guess the values from the box and whisker plots in Figures 2 and 3. I ask the authors to create one or more tables showing the means, ± a suitable measure of variance, for the variables measured, organised by sex, season and site.
AR: Done. We have added a table as supplementary material (Supplementary Table S1) with descriptive statistics (mean, standard deviation, SD, and sample size, N) for the variables describing nest use (number of nests per individual, number of times that each nest was used, distance between consecutive nests in m), their spatial spread (MCP, minimum convex polygon, ha) and distance (D. in m) to some artificial structures.
6) The box and whisker plots (Figures 2 and 3) have no explanation of what they are showing (presumably they are means ± some measure of variance).
AR: Done. We have added the following text, to clarify the information presented in these figures: “Typical boxplot structure (median, lower and upper quartiles, and whiskers calculated as 1.5 times the interquartile range) is shown together with the actual values of each variable (small circles).”
7) Line 255: Is a p value of 0.0898 marginally significant? I think not. Normally figures >0.05 are not considered significant. Also, looking at Tables 1 & 2, there are several values >0.08 that are also reported to be marginally significant. Are these errors or can this be explained?
AR: A marginally significant test result refers to a statistical test that produces a p-value that is close to the pre-determined threshold for statistical significance, typically set at 0.05. It is a common practice to consider p-values between 0.05 and 0.10 as marginally significant. As the 0.05 significance threshold is arbitrarily determined, discarding results with p-values somewhat above this threshold may led to overlook marginally significant findings and may be limiting the overall understanding of a research question. To clarify this issue, we have included at the end of the last paragraph of the Statistical Analyses section the following sentence “The significance level for statistical analyses was set at p=0.05, and a marginally significant result was defined as 0.05 < p < 0.1.”.
There was an error in the text of figures 2 and 3 that in the first version said, “Only variables that showed a significant effect of season or sex, as determined by GLM or GLMM (in Table 1 for the university nests), are presented.”. This was an error and has been corrected in the revised version to “significant or marginal significant effect” because some tests for some variables (as MCP and distance to road sin the university for instance in Figure 2) were marginally significant.
Although the paper is generally well-written, I have listed a substantial number of small edits, corrections and suggested improvements to the English which should be carried out before the paper can be published. I trust that this is helpful because it has taken a lot of time.
Recommended changes to the text:
Lines 13-15: I suggest these two sentences are re-written to e.g. ‘Our results show that there is a sex difference in the structures selected for nesting. Males used a greater variety of structures than females which were more selective and showed a preference for certain structures that offered a higher degree of shelter. Both sexes used a similar number of nests, but males’ nests were more widely dispersed and they changed nests more often than females which re-used the same nest more frequently.’
AR: Done
Line 30: ‘nesting hedgehog ecology’ should read ‘hedgehog nesting ecology’
AR: Done
Line 32: I suggest a re-write to: ‘Further studies of nest location at a micro habitat level are needed to contribute to a better understanding ..’
AR: Done
Line 53: ‘making winter nest’ should be ‘making winter nests’
AR: Done
Lines 54 to 55: ‘but however’ should be ‘however’
AR: Done
Lines 55 to 56: ‘hibernation is reduced up to two months’ should be ‘hibernation is reduced by up to two months’
AR: Done
Lines 62 to 63: ‘choosing wisely the day nest location’ should be ‘a wise choice of day nest location’
AR: Done
Line 65: ‘European hedgehog’ should be ‘the European hedgehog’
AR: Done
Line 67: ‘and urbanisation process.’ should be ‘and urbanisation.’ (urbanisation is a process)
AR: Done
Line 71 to 72: ‘the continuous growing of cities’ should be ‘the constant growth of cities’
AR: Done
Line 74: ‘play then’ should be ‘then play’
AR: Done
Line 78: ‘by hedgehog’ should be ‘by hedgehogs’
AR: Done
Line 80: delete ‘in them’
AR: Done
Line 85: ‘of intensively‘ should be ‘of an intensively’
AR: Done
Lines 88 to 89: ‘the structures selected to locate the nests’ should be ‘the structures in which nests were located’
AR: Done
Line 89: ‘nests’ should be ‘nests’ ’
AR: Done
Lines 98 to 99: ‘good shelters along the place, Mixed between pavement’ should be ‘good shelters between areas of pavement and wide-open grass’
AR: Done
Line 132: the initial hedgehog. Does this mean that one hedgehog was tagged for a trial period of three days at the beginning of the study? If so, this would be better as ‘a hedgehog was tagged for a trial period of three days.’
AR: Yes, exactly. We have changed the sentence to clarify it. Thank you.
Line 135: ‘to relocate it’ should be ‘in relocating it.’
AR: Done
Line 136: ‘that continuously broadcast signal’ should be ‘that continuously broadcast a signal’
AR: Done
Line 147: ‘GPS were’ should be ‘GPS devices were’
AR: Done
Line 148: ‘the sunset’ should be ‘sunset’
AR: Done
Line 159: ‘if hedgehog’ should be ‘if a hedgehog’
AR: Done
Lines 160 to 161: ‘in four different categories depending on the degree of consistency’ should be ‘into four structural categories’
AR: Done
Lines 174-178: avoid using an upper-case E for e.g.
AR: Done
Line 180: delete ‘different’ (of course the categories are different!)
AR: Done
Line 187: ‘absence when’ should be ‘absence was recorded when’
AR: Done
Line 195: ‘distance to the feeders were used’ should be ‘distance to the feeders was used’
AR: Done
Lines 199 -200: Delete ‘Measures of habitat characterization were taken for all the nests located.’ Alter the next sentence to ‘Habitat characteristics were measured for all the nests located, using a 5-metre radius around the central point of each nest.’
AR: Done
Line 210: change ‘if’ to ‘whether’ – it is not wrong but whether sounds better.
AR: Done
Line 215: ‘data of the university campus’ should be ‘data from the university campus’
AR: Done
Line 216: ‘data of spring-summer’ should be ‘data from spring-summer’
AR: Done
Line 216: ‘data of the same hedgehog’ should be ‘data from the same hedgehog’
AR: Done
Lines 218-219: delete ‘with’ to read ‘may contribute data for both seasons,’
AR: Done
Lines 219-220: Alter sentence to ‘….seasons, and in the urban forest, models for nest distances to artificial structures and times occupied.’
AR: Done
Lines 221-222 alter sentence to read ‘… contributed just once, GLM was used.’
AR: Done
Line 229: ‘MCPs’ remove the s
AR: Done
Line 233-234: this should be altered to read ‘FAMD allows continuous and categorical data to be integrated in the same analysis. Thus it allowed us to investigate….’
AR: Done
Line 236: replace ‘relative’ with ‘relating’
AR: Done
Line 237: replace ‘FMAD were performed using the R package…’ with ‘We used the R package…’
AR: Done
Line 239-241: The meaning is a little unclear and (assuming I understand this properly) you may wish to edit these lines to read ‘… dimension using correlation in the case of continuous variables, and the significance and R2 of an ANOVA when testing for differences of coordinates between levels of each categorical variable [31].’
AR: Done
Line251: change ‘The results of the university campus . . .’ to ‘The results from the university campus . . . ‘
AR: Done
Line 263: ‘nest’ should be ‘nests’
AR: Done
Line 277: ‘relative ‘ should be ‘relating’
AR: Done
Line 278: ‘sign’ should be ‘signs’
AR: Done
Lines 284 and 285: ‘relative ‘ should be ‘relating’
AR: Done
Line 286: ‘sign’ should be ‘signs’
AR: Done
Table 2: The GLMM result for ‘Times occupied’ is shown as 0,0017 when it should be 0.0017
AR: Done
Line 300: ‘a lower number of’ can be replaced with ‘fewer’
AR: Done
Line 318: is a slope of 25° a mean? This is vaguely written.
AR: Done. We have rewritten the sentence clarifying it.
Table 3 & Table 4 and also the second graph in Figure 4: note incorrect spelling of Herbaceous
AR: We have corrected the spelling errors in Tables 3 and 4 and also in the second graph in Figure 4. Thank you!
Line 365: delete ‘The’
AR: Done
Line 367: insert ‘The’ to read ‘The first dimension. . . ’
AR: Done
Line 371: insert ‘The’ to read ‘The second dimension. . . ’
AR: Done
Line 387: insert ‘the’ to read ‘. . . at the university . . . ’
AR: Done
Line 388: ‘at them.’ should be ‘in them.’
AR: Done
Line 388: Nest at university’ should be ‘Nests at the university’
AR: Done
Line 426: ‘the most important correlations’ these are presumably all statistically significant with a p<0.05? If so, say so.
AR: This is not the case because, with the high number of nests used to calculate the correlations (130), any correlation larger than 0.17 will be considered significant with a p-value < 0.05. In other words, variables explaining approximately 3% of the variance in the coordinates of a given dimension are considered significant at this p-value. It is worth noting that frequently these low yet significant correlations can be attributed to a few extreme values. Instead, we visually inspected the highest correlation for each dimension and presented their scatterplots in Supplementary Figure 3. This enables readers to easily assess the strength of the relationship.
This visual inspection led us to choose correlations that, in all cases, are > 0.45, meaning that the variable explains at least 20% of the variance in the dimension coordinates. Therefore, our selection of the variables used to interpret dimensions was based on effect size (R2 > 0.2). The only exception was the correlation of rocks with dimension 7. However, this high correlation was caused by an outlier, so we did not interpret this dimension.
Line 458: Although at p=0.0898 this was not quite significant? See my comment 7) above.
AR: We considered this p-value as marginally significant, as explain in the response to comment number 7.
Line 466: ‘what’ should be ‘that’
AR: We have rewritten the whole sentence to: “The combination of both results together shows that the exploratory behaviour of males may be staggered. Males do not change areas drastically from one night to the next (distance between consecutive nests is similar to that of females), but it is the cumulative effect over several nights that contributes to larger home range areas for males.”, so what has disappeared in the new text.
Lines 458 & 467 how much larger/higher?
AR: Done. We have rewritten the sentence to reflect that information.
Lines 515 & 518: ‘pampa’ should be ‘pampas’
AR: Done
Lines 515-516 ‘Both plant species share in common that, unlike many shrubs, they are difficult-to-access structures’ can be edited down to ‘Both plant species, unlike many shrubs, are difficult-to-access structures,’
AR: Done
Line 519: ‘at’ should be ‘in’
AR: Done
Line 526: it would be clearer to change ‘besides’ to ‘as well as’
AR: Done
Line 534: ‘the 26.3% of the cases’ should be ‘in 26.3% of cases’
AR: Done
Line 536-537: ‘At the urban forest, 93.3% of the times the secondary structure . . . ‘‘ should be ‘In the urban forest, for 93.3% of nests the secondary structure . . .
AR: Done
Line 538: ‘was’ should be ‘nests were’
AR: We agree that the sentence was somewhat confusing, which we believe led to a proposed change that altered the meaning of what we meant. For this reason, we have decided to rewrite the sentence to make it clearer: “In the urban forest, for 93.3% of nests, the secondary structure was composed of live plants, mainly Brachypodium retusum, while at the university campus, it was predominantly composed of leaf litter (66.7% of the cases)”
Line 539: ‘Therefore, in the university campus leaf litter seems to contribute to increase suitability of . . .’ should be ‘Therefore, at the university campus, leaf litter seems to contribute to an increased suitability of . . . ‘
AR: Done
Line 543: ‘likely contributes to decrease their habitat quality’ change to ‘is likely to contribute to a decreased habitat quality’
AR: Done
Line 550: ‘likely more’ should be ‘likely to be more’
AR: Done
Line 555: ‘likely found’ should be ‘likely to be found’
AR: Done
Line 570: ‘in a mountain’ should be ‘on a mountain’ and ‘affect to the cost’ should be ‘affect the cost’ and ‘contribute’ should be ‘help’
AR: Done
Line 572: ’showed lower number of nests’ is better as ‘used fewer nests’
AR: Done
Line 574: ‘less costly’ should be ‘at a lower cost’
AR: Done
Line 576-578: To avoid clumsy brackets this could perhaps be re written as ‘Nests of both sexes were closer to the five available cat feeders in the urban forest but were more distant in the university campus with 14 feeders.’ Question - are the feeders more widely dispersed on the campus?
AR: Done.
In response to the question about the spatial distribution of the feeders, we have decided to modify the original map of the study areas to include the locations of the cat feeders. We hope that help clarify these doubts.
Line 580: ‘to this difference’. Which difference? This is a new paragraph and it is unclear which of the differences is now being discussed.
AR: Thank you for the suggestion. We have modified the sentence in the text to make it clearer that we are referring to the fact that the different management of the urban parks could have influenced the dependency of the hedgehogs on cat feeders in the urban forest. We hope the new wording helps to clarify it.
Line 583: ‘contribute’ should be ‘help’
AR: Done
Line 589: ‘make more likely’ should be ‘make it more likely’ and ‘be’ should be ‘will be’
AR: Done
Line 608: presumably this should say ‘plastics and chocolate wrappers; found in 26.7% of nests.’
AR: Done
Line 609: ‘reaffirm’ should be ‘reaffirms’
AR: Done
Line 614: ‘with presence’ should be ‘with the presence’
AR: Done
Line 615: delete unnecessary words ‘in space’
AR: Done
Line 616: change ‘support’ to ‘suggest’ or ‘indicate’
AR: Done
Line 618: ‘distance to’ should be ‘distance of nests to’
AR: Done
Lines 621-623: Edit this sentence from ‘Opposite results were found at the urban forest, where the presence of a road running through the center made females be closer to roads than males, evidencing that the distribution of elements in urban areas is important’ to ‘Opposite results were found in the urban forest, where a road ran through the center, and female nests were closer to roads than those of males; evidence that the distribution of elements in urban areas is important’
AR: Done
Lines 624-625: ‘dependency of hedgehogs for supplementary food, having at the urban forest the smallest distances to the cat feeders from the nest.’ would be better as ‘the dependence of hedgehogs on supplementary food with the nests closer to the cat feeders in the urban forest.’
AR: Done
Line 629: ‘contrast’ should be ‘contrasts’
AR: Done
Line 633: ‘Difference’ should be ‘Differences’ and ‘likely’ should be ‘are likely to be’
AR: Done
Line 637: change ‘this species’ to ‘hedgehogs’ to avoid repetition of ‘this species’ on the next line.
AR: Done

Reviewer 2 Report
1. I believe that the area chosen for the study is not representative of the southeast region of Spain in the title and suggest that the title be changed to better fit the findings of this study.
2. It is proposed to add a description of the population size of the study species in the study area, as well as its distribution in southeast Spain.
3. Lines 60-62 lack citation support for the description of the hedgehog's vulnerability to sleeping during the day.
4. In 2.1 you should first explain why the place you have chosen for your study is representative of southeast Spain.
5. Cat feeders on campus have an impact on the energy intake strategies of hedgehogs over winter and even throughout the year, and different plants at the micro-landscape scale do not directly account for their nesting site preferences in different seasons, as there are too many anthropogenic factors throughout the year.
6. In 2.3 As different seasons will have different sunrise and sunset times, it is recommended that you state in the text that your recording times will be adjusted for the different seasons to make the research data more meaningful.
Author Response
Response to Reviewer 2 Comments
Comments and Suggestions for Authors:
- I believe that the area chosen for the study is not representative of the southeast region of Spain in the title and suggest that the title be changed to better fit the findings of this study.
AR: We agree with the reviewer suggestion, so we have updated the tittle of the article to “Nesting ecology of European hedgehogs (Erinaceus europaeus) in urban areas in Southeast Spain: nest habitat use and characteristics”
- It is proposed to add a description of the population size of the study species in the study area, as well as its distribution in southeast Spain.
AR: Done. We added the number of adult hedgehogs monitored for each population during the study period.
Regarding the distribution of the European hedgehog in the southeast of Spain, we cannot provide detailed data beyond saying that it is widely distributed, as we have commented in the introduction, due to the lack of detailed studies of this species in the country.
- Lines 60-62 lack citation support for the description of the hedgehog's vulnerability to sleeping during the day.
AR: Done. We have added a reference supporting the known sleeping behaviour of the hedgehogs, and we have modified the next sentence to not make it look so blunt.
- In 2.1 you should first explain why the place you have chosen for your study is representative of southeast Spain.
AR: We added a sentence at the beginning of the paragraph 2.1 to first explain why we have chosen these two study sites, following the reviewer suggestion: “We selected two areas representative of two types of urban habitats used by hedgehogs in the region.”
- Cat feeders on campus have an impact on the energy intake strategies of hedgehogs over winter and even throughout the year, and different plants at the micro-landscape scale do not directly account for their nesting site preferences in different seasons, as there are too many anthropogenic factors throughout the year.
AR: Sorry, we don't quite understand what the reviewer is referring to and what we should do about it. We agree that the feeders have a primary effect on the energy intake strategies of hedgehogs over winter and even throughout the year, as we already emphasized in lines 604-607. For this reason, the distance from the resting nests to the feeders has been taken into account as a variable that influences the establishment of nests (lines 206-212) (Note: line numbers refer to manuscript version with changes accepted).
In any case, and in order to highlight the effect of feeders on the spatial distribution of hedgehogs’ nests, we have decided to modify the map in Figure 1 to show their location in the two study areas. We hope it helps to clarify that in both areas, the feeders are widely distributed throughout the space.
- In 2.3 As different seasons will have different sunrise and sunset times, it is recommended that you state in the text that your recording times will be adjusted for the different seasons to make the research data more meaningful.
AR: Done. We have added a sentence explaining this, following the reviewer's suggestion.
Reviewer 3 Report
Introduction – gives a good overview about biology and conservation of the species and describes the aims quit well
Description of the study area is okay, Cat feeder as a habitat feature should be considered in 2.4.1
Fig 1: Maps should be marked with a), b), c) d): I think the left bottom map is not necessary. Description of blue polygons (I guess water) is missing.
Capture procedure and GPS device are clear to me. Is it right that you found the nests after removing the logger and analysing the data?
2.4.1 Description and complementary structure is quite hard to read. Maybe it is possible to create a table and describe it there. Also, artificial structures and habitats could be structures clearer e.g. by bullet points
Table 1 and 2: Marginally significant means 0,05<p<0,1? If yes, please describe it. Normally only p value < 0,05 is considered.
Figure 1: Meaning of boxes, bares and dots should be described in the description. I find the different colours for the sexes confusing. Maybe there is a better solution. In the figure also marginally significant results are shown? These results are not significant by definition.
Distance to artificial structures: Did you also consider the availability of these structures? I guess at the University campus streetlight is much nearer that at the forest in general. So, did the hedgehogs place the nest nearer to streetlight or was there no other chance?
3.3. Nest type could be presented as a figure and table 3 should be shifted to an attachment of supplemental material
Discussion and conclusions are well structured and clear to me.
Author Response
Response to Reviewer 3 Comments
Comments and Suggestions for Authors:
Introduction – gives a good overview about biology and conservation of the species and describes the aims quit well.
Description of the study area is okay, Cat feeder as a habitat feature should be considered in 2.4.1
AR: Cat feeders as a habitat feature is considered in 2.4.1, in lines 206-2012 (Note: line numbers refer to manuscript version with changes accepted).
Likewise, and thanks to the reviewer's suggestion, we wanted to highlight its importance and for this reason we have modified Figure 1 to show the location of the feeders on the map in each of the study areas.
Fig 1: Maps should be marked with a), b), c) d): I think the left bottom map is not necessary. Description of blue polygons (I guess water) is missing.
AR: Done. We have improved the maps shown in Figure 1 and, following the reviewer's recommendation, we have numbered both study areas to make them easier to locate. On the other hand, the authors consider that the detailed location of both study areas on the coast of Alicante is useful for readers unfamiliar with the area, so we have preferred to leave the detailed map (left bottom).
Finally, we have updated the map description to include that the blue polygons refer to water.
Capture procedure and GPS device are clear to me. Is it right that you found the nests after removing the logger and analysing the data?
AR: Yes, that’s correct. We could only find the nests once we had access to the data collected in the GPS devices, and for that we needed to recover the GPS devices first. They did not send us the information remotely.
2.4.1 Description and complementary structure is quite hard to read. Maybe it is possible to create a table and describe it there. Also, artificial structures and habitats could be structures clearer e.g. by bullet points
AR: We agree with the reviewer that this part might be hard to read, but due to the larger number of tables and figures already presented in the manuscript we are reluctant to add new ones. We sincerely hope that the photographs provided in Supplementary Figure 2 will be helpful in clarifying the possible doubts of the reader regarding the description of the structures.
Table 1 and 2: Marginally significant means 0,05<p<0,1? If yes, please describe it. Normally only p value < 0,05 is considered.
AR: To clarify this issue we have included at the end of the last paragraph of the Statistical Analyses section the following sentence “The significance level for statistical analyses was set at p=0.05, and a marginally significant result was defined as 0.05 < p < 0.1.”.
Figure 1: Meaning of boxes, bares and dots should be described in the description. I find the different colours for the sexes confusing. Maybe there is a better solution. In the figure also marginally significant results are shown? These results are not significant by definition.
AR: Done. We have corrected the description of the figure.
Regarding the colors of the last MCP in Figure 2, the authors chose to change the colors of the boxes of the last graphic to highlight that the results shown are only from the urban forest hedgehog population.
There was an error in the text of figures 2 and 3 that in the first version said, “Only variables that showed a significant effect of season or sex, as determined by GLM or GLMM (in Table 1 for the university nests), are presented.”. This was an error and has been corrected in the revised version to “significant or marginal significant effect” because some tests for some variables (as MCP and distance to road in the university for instance in Figure 2) were marginally significant.
Distance to artificial structures: Did you also consider the availability of these structures? I guess at the University campus streetlight is much nearer that at the forest in general. So, did the hedgehogs place the nest nearer to streetlight or was there no other chance?
AR: We are aware that the availability of artificial structures is different between our study areas, but we have not quantified formally their availability. In the case of streetlights, the university campus is better illuminated than the urban forest, and this explains the shorter distances to them at the university. However, the interesting result is that in both sites female nests were located closer to the streetlights compared to male nests, as we highlight and discuss in lines 329-330 and 593-597. Within each study site the availability of artificial structures is the same for both sexes, thus the detected differences between them point to a different selection process of males and females.
3.3. Nest type could be presented as a figure and table 3 should be shifted to an attachment of supplemental material.
AR: Thanks a lot for the suggestion. However, due to the large number of figures and tables that we already included in the article, the authors prefer to leave only Table 3 with all the information, instead of creating a new table and a new figure from it.
Discussion and conclusions are well structured and clear to me.
Reviewer 4 Report
Summary: The paper presents a survey of hedgehog nesting sites in two urban areas in southeastern Spain. Male hedgehogs are shown to use more varied nest structures than females and to nest over a wider area. Female hedgehogs were shown to be more likely to re-use nests. Nest location in the two areas was influenced by both topography and management. The paper presents novel information on detailed environmental characteristics chosen by hedgehogs und suggests that any future studies should focus on nest micro habitat.
General Comments:
Manuscript is well written, relevant to the journal and sensibly structured. Sources cited are all relevant to the current study with several recent papers on the study organism of particular importance. The introduction gives enough general and more specific background to explain and justify the present study and identifies gaps in current knowledge.
Suitable level of detail is presented in the materials section, on the study areas, animal handling and tracking, nest and habitat monitoring and statistical analyses (but see specific comments below). Enough detail to allow repetition of the study. Images/information in the supplementary material will aid in nest classification. Experimental design is robust, with decent sample sizes obtained, and the range of data the authors have collected are impressive.
Results are clearly displayed in tables and figures, although I wonder if some of the supplementary material might be added for greater clarity. I do have an issue with the highlighting of results/p-values above 0.05 as significant in tables 1 and 2, and the related use of "marginally significant" in text (see specific comments).
I am impressed with the range of analyses that the authors have conducted, they have certainly made the most of their data sets. All of the analytical approaches and decisions made during analysis are clearly justified.
The discussion section is on the long side, but does cover a lot of relevant material and relates back to the results well. Length/detail of the discussion is justified by the range of data collected and presented. All aspects of the study are given suitable coverage in the discussion, although there are some sections that might be edited with no loss of meaning.
Specific Comments:
Line 56 - Re-word to "reduced to two months"
Line 73 - citation for UN population estimate?
Lines 159/160 - "Only if hedgehog..." is not a sentence and meaning needs to be clarified.
Line 189 - define what "CNIG" stands for
Lines 255, 301, 303, 308 - use of "marginally significant" and related p-values in tables 1 and 2 more than 0.05 but highlighted in bold. If using the accepted values for significance then these results are not significant and should be treated/described accordingly.
Overall the manuscript is addressing a novel aspect of a well-known and much loved organism and the results will be of interest to the scintific community and the wider public. The article is well written and scientifically sound, advances our knowedge of the organism and I will be happy to see it published.
Some badly worded phrases throughout, but nothing serious enough to cause any misunderstanding. Only minor edits required.
Author Response
"Please see the attachment."

Reviewer 5 Report
These authors present an interesting and well-designed study of hedgehogs' nesting habits and the differences between males and females regarding this matter. In general, the manuscript is well-written and there are no major concerns regarding it. However, I sincerely believe authors can improve their manuscript significantly if they try to (1) provide more references at the beginning of their discussion. Most of the first part of your discussion is a personal interpretation of your results and very few references are cited. Even though I honestly agree with the authors' interpretation and thoughts, I think they should provide references, perhaps regarding other species to support their thoughts. Moreover, authors should try to give (2) a more practical perspective to their work. I certainly agree that your work gives important information of hedgehog behaviour and ecology and it is a relevant contribution to understanding the species and help to its conservation. However, as far as I know, hedgehogs' conservation status in the Iberian peninsula is not comparable to the UK (for instance). The stability of the population is completely different due to the absence of predators. Nevertheless, it is certainly important to understand how the proximity to humans is influencing their behaviours.... what this means under an One Health perspective or infectious diseases. Thus, your study gives a contribution to something more than merely understanding hedgehogs in a particular area. I would suggest the authors to also give a short perspective regarding this.
Here I also present other particular details to be corrected in your manuscript.
L50-51 - The distribution of the European hedgehog is not 100% correct. This species can also be found in the Azores islands and New Zealand so the way you describe the distribution is not accurate.
L57 - In southern Portugal the hibernation has also disappeared for some hedgehogs.
L241 - please indicate the critical value of P considered in this ANOVA testing or other statistical inferences.
"Females nested closer to paths and streetlights at the university while males were closer to roads" - I believe this is can also be related to the fact that male hedgehogs are more often victims of car collisions than females. Some authors have identified this with rescue centres data and I believe this is also important to be mentioned in your manuscript because this habit can explain the differences in hedgehogs' mortality
Author Response
Response to Reviewer 5 Comments
Comments and Suggestions for Authors:
These authors present an interesting and well-designed study of hedgehogs' nesting habits and the differences between males and females regarding this matter. In general, the manuscript is well-written and there are no major concerns regarding it. However, I sincerely believe authors can improve their manuscript significantly if they try to (1) provide more references at the beginning of their discussion. Most of the first part of your discussion is a personal interpretation of your results and very few references are cited. Even though I honestly agree with the authors' interpretation and thoughts, I think they should provide references, perhaps regarding other species to support their thoughts. Moreover, authors should try to give (2) a more practical perspective to their work.
I certainly agree that your work gives important information of hedgehog behaviour and ecology and it is a relevant contribution to understanding the species and help to its conservation. However, as far as I know, hedgehogs' conservation status in the Iberian peninsula is not comparable to the UK (for instance). The stability of the population is completely different due to the absence of predators. Nevertheless, it is certainly important to understand how the proximity to humans is influencing their behaviours.... what this means under an One Health perspective or infectious diseases. Thus, your study gives a contribution to something more than merely understanding hedgehogs in a particular area. I would suggest the authors to also give a short perspective regarding this.
AR: Done. We have added more references in the discussion part to support our conclusions and we have added some management recommendations to give a more practical perspective to our work, following the reviewer's suggestions. However, and despite the interest of the One Health approach, we have not addressed in in this article as we believe that the data provided is not sufficient to draw solid or useful conclusions on this topic.
Here I also present other particular details to be corrected in your manuscript.
L50-51 - The distribution of the European hedgehog is not 100% correct. This species can also be found in the Azores islands and New Zealand so the way you describe the distribution is not accurate.
AR: Done. We have modified the description of the distribution, by mentioning only the natural distribution of the species, to make it more accurate.
L57 - In southern Portugal the hibernation has also disappeared for some hedgehogs.
AR: The authors appreciate this information but unfortunately, we have not been able to find any reference that confirms this information. For that reason, we have preferred not to modify the sentence.
L241 - please indicate the critical value of P considered in this ANOVA testing or other statistical inferences.
AR: To clarify this issue we have included at the end of the last paragraph of the Statistical Analyses section the following sentence “The significance level for statistical analyses was set at p=0.05, and a marginally significant result was defined as 0.05 < p < 0.1.”.
"Females nested closer to paths and streetlights at the university while males were closer to roads" - I believe this is can also be related to the fact that male hedgehogs are more often victims of car collisions than females. Some authors have identified this with rescue centres data and I believe this is also important to be mentioned in your manuscript because this habit can explain the differences in hedgehogs' mortality
AR: The authors partially agree with the reviewer's comment. We agree, as many studies have shown, that males are more likely to be victims of roadkill than females. However, we do not consider nest location as a direct cause, but the exploratory behavior of the males, which leads them to travel greater distances in search of females.
The location of the nests is, in this case, another consequence of this exploratory behavior, which leads the males of the university campus to approach the periphery of the study area, that is surrounded by roads. Contrary, is the same behavior that causes the opposite effect at the urban forest, were males located their nests farther from roads, due to the location of a road that runs through the center of the area, as we discussed in the discussion.
We therefore believe that the data we provide in this study are not sufficient to relate them to the car accidents males could have. So, we have limited ourselves to suggesting corrective measures must be implemented when suitable nesting sites are found particularly close to the roads (see “Conclusions”).